# Major trauma presentations and patient outcomes in English hospitals during the COVID-19 pandemic: An observational cohort study

Carl Marincowitz[1]*, Omar Bouamra[2], Tim Coats[3], Dhushy Surendra Kumar[4], David Lockey[5,6], Lyndon Mason[7], Virginia Newcombe[8], Julian Thompson[9], Antoinette Edwards[2], Fiona Lecky[1,2]

1 Centre for Urgent and Emergency Care Research (CURE), Health Services Research School of Health and Related Research, University of Sheffield, Sheffield, United Kingdom, 2 Trauma Audit Research Network, University of Manchester, Manchester, United Kingdom, 3 Emergency Medicine Academic Group, Department of Cardiovascular Sciences, University of Leicester, Leicester, United Kingdom, 4 Department of Critical Care, Anaesthesia and Pre-hospital Emergency Medicine, University Hospital Coventry, Coventry, United Kingdom, 5 London's Air Ambulance, Royal London Hospital, London, United Kingdom, 6 North Bristol NHS Trust, Bristol, United Kingdom, 7 Liverpool University Hospitals NHS Foundation Trust, University of Liverpool, Liverpool, United Kingdom, 8 Division of Anaesthesia, University of Cambridge, Cambridge, United Kingdom, 9 Department of Anaesthesia and Intensive Care Medicine, Southmead Hospital Intensive Care Unit, Southmead Hospital, North Bristol NHS Trust, Bristol, United Kingdom

* C.Marincowitz@Sheffield.ac.uk

**Data Availability Statement:** The data used for this study were collected by the Trauma Audit and Research Network (TARN), based at the University

## Abstract

### Background

Single-centre studies suggest that successive Coronavirus Disease 2019 (COVID-19)-related "lockdown" restrictions in England may have led to significant changes in the characteristics of major trauma patients. There is also evidence from other countries that diversion of intensive care capacity and other healthcare resources to treating patients with COVID-19 may have impacted on outcomes for major trauma patients. We aimed to assess the impact of the COVID-19 pandemic on the number, characteristics, care pathways, and outcomes of major trauma patients presenting to hospitals in England.

### Methods and findings

We completed an observational cohort study and interrupted time series analysis including all patients eligible for inclusion in England in the national clinical audit for major trauma presenting between 1 January 2017 and 31 of August 2021 (354,202 patients). Demographic characteristics (age, sex, physiology, and injury severity) and clinical pathways of major trauma patients in the first lockdown (17,510 patients) and second lockdown (38,262 patients) were compared to pre-COVID-19 periods in 2018 to 2019 (comparator period 1: 22,243 patients; comparator period 2: 18,099 patients). Discontinuities in trends for weekly estimated excess survival rate were estimated when lockdown measures were introduced using segmented linear regression.

of Manchester and analysed by the TARN research committee. The data used for this study can be requested from the TARN research committee (https://www.tarn.ac.uk/Content.aspx?ca=9&c=3810 research@tarn.ac.uk.) but may require a data sharing agreement and additional ethics approvals.

**Funding:** This study did not receive any external funding and has been completed by the Trauma Audit and Research Network (TARN) research committee and collaborators.

**Competing interests:** The authors have declared that no competing interests exist.

**Abbreviations:** COVID-19, Coronavirus Disease 2019; ED, emergency department; ICU, intensive care unit; ISS, Injury Severity Score; ITS, interrupted time series; MTC, major trauma centre; TARN, Trauma Audit and Research Network.

The first lockdown had a larger associated reduction in numbers of major trauma patients (−4,733 (21%)) compared to the pre-COVID period than the second lockdown (−2,754 (6.7%)). The largest reductions observed were in numbers of people injured in road traffic collisions excepting cyclists where numbers increased. During the second lockdown, there were increases in the numbers of people injured aged 65 and over (665 (3%)) and 85 and over (828 (9.3%)).

In the second week of March 2020, there was a reduction in level of major trauma excess survival rate (−1.71%; 95% CI: −2.76% to −0.66%) associated with the first lockdown. This was followed by a weekly trend of improving survival until the lifting of restrictions in July 2020 (0.25; 95% CI: 0.14 to 0.35). Limitations include eligibility criteria for inclusion to the audit and COVID status of patients not being recorded.

## Conclusions

This national evaluation of the impact of COVID on major trauma presentations to English hospitals has observed important public health findings: The large reduction in overall numbers injured has been primarily driven by reductions in road traffic collisions, while numbers of older people injured at home increased over the second lockdown. Future research is needed to better understand the initial reduction in likelihood of survival after major trauma observed with the implementation of the first lockdown.

## Author summary

### Why was this study done?

- Previous studies assessing the impact of COVID lockdowns on major trauma have used data collected at single or a small number of hospitals.

- There is conflicting evidence regarding whether changes to health services during the pandemic led to worse outcome for patients following major trauma.

### What did the researchers do and find?

- We used national registry data for all trauma-receiving hospitals in England to assess the impact of successive COVID lockdowns on the characteristics, care pathways, and outcomes of major trauma patients.

- We used a quasi-experimental method that accounted for changes in the characteristics of major trauma patients to identify changes in likelihood of survival when lockdown measures were introduced.

- We found large reductions in major trauma during both lockdowns, particularly related to road traffic collisions. Reductions were smaller for patients aged over 65.

- Likelihood of survival following major trauma fell when the lockdown measures were first introduced. Reassuringly, likelihood of survival returned to prelockdown levels over the period of the first lockdown.

**What do these findings mean?**

- Falls from standing is the most common cause of major trauma in older adults. Lockdown restrictions appear not to have not have altered likelihood of such trauma.

- Disruption of services due to COVID restrictions may have reduced likelihood of survival during the initial part of the first COVID wave in England.

- This study only included trauma patients who attended hospital and met national audit inclusion criteria and COVID status of patients was not included in analysis.

## Introduction

To control transmission of the Coronavirus Disease 2019 (COVID-19) during the pandemic, the United Kingdom government implemented successive lockdown measures in England [1]. The first lockdown was announced on 23 March 2020, and, following a period of relaxation, a second lockdown was announced on 30 October due to the emergence of the Alpha variant. By September 2021, there was a 16% reduction in road traffic from prepandemic levels [2]. This presents a unique opportunity to assess the impact of potential road traffic reducing public health measures on major trauma [3].

There is some evidence that restrictions associated with lockdowns may have contributed to increased nonaccidental injury, domestic violence, and self-harm related to deteriorating mental health [4–6]. Internationally, there is also evidence that the diversion of healthcare resources to treating patients with COVID-19, particularly intensive care capacity, may have led to worse outcomes for patients presenting with major trauma [7].

A recent systematic review including 35 studies from 14 countries assessing the impact of lockdown on major trauma admissions and outcomes in the first COVID wave in predominantly single-centre studies found that lockdown periods were associated with reductions in admissions for major trauma, particularly related to motor vehicles [8]. The review found a significant increase in major trauma related to self-harm and firearms. The authors also reported no change in pooled risk of death during lockdown periods [8]. In the UK, there have been single-centre or regional assessments of the impact of the first lockdown on the numbers and characteristics of major trauma patient presentations [9–12]. There has, however, been no previous evaluation at a national level in England.

We therefore aimed to describe the impact of COVID-19 including successive lockdowns on the number, demographics, injury mechanism, severity, care pathways, and outcomes of patients with major trauma presenting to hospitals in England. We specifically tested the hypothesis that disruption of care for major trauma patients reduced the likelihood of survival.

## Methods

We conducted an observational cohort study and interrupted time series (ITS) analysis to test that hypothesis that disruption of care due to COVID restrictions reduced likelihood of survival following major trauma. The protocol and prespecified analysis are publicly available and included in S1 Appendix [13]. All analyses were conducted in accordance the prespecified

plan with no additional exploratory analysis. This study is reported as per the Strengthening the Reporting of Observational Studies in Epidemiology (STROBE) guideline (S1 Checklist).

## Data set

Routinely collected patient data from all NHS England trauma-receiving hospitals submitted to the national clinical audit for major trauma—the Trauma Audit and Research Network (TARN). We studied patients presenting between 1 January 2017 and 31 August 2021. All trauma-receiving hospitals (major trauma centre (MTC) and trauma units) in England submit data on eligible trauma patients to the TARN database for the purposes of audit, governance, research, and benchmarking.

The TARN database includes the anonymised records of patients of any age who sustain injury resulting in hospital admission >72 hours, critical care admission, transfer to a tertiary/specialist centre, or death within 30 days. Isolated femoral neck or single pubic ramus fractures in patients >65 years and simple isolated injuries are excluded. After study inclusion, a dataset of prospectively recorded variables covering demographics plus injury-related physiological, investigation, treatment, and outcome parameters are collated using a standard web-based case record form by TARN hospital audit coordinators. Injury descriptions from imaging, operative, and necropsy reports are submitted by TARN coordinators—all injuries are coded centrally using the Abbreviated Injury Scale; this enables calculation of the Injury Severity Score (ISS) [14].

## Analyses

To illustrate changes in the number and mechanisms of injury associated with successive lockdowns, a quarterly time series analysis for the period 1 January 2017 to 31 August 2021 was conducted for the total numbers of major trauma patients in England. The time series was further stratified by management in an MTC, road traffic collisions, intentional injuries, falls, and sport.

Demographic characteristics of major trauma patients including age, sex, physiology, injury severity, and body region injury for the first lockdown (24 March to 3 July 2020 inclusive) and second lockdown (1 November 2020 to 16 May 2021 inclusive) were compared to equivalent pre-COVID-19 periods in 2018 to 2019. Similarly, to assess changes in management pathways for patients, the total and proportion of traumatically injured patients who were received by or transferred to an MTC, assessed by a consultant in the emergency department (ED), received CT imaging, undergoing operative intervention, admitted to critical care, and dying before discharge or 30 days post injury (whichever was earlier) were compared in equivalent pre-COVID and lockdown time periods. Absolute changes with 95% confidence intervals were calculated for the population and care pathway characteristics assessed.

To specifically assess if there were any changes in risk-adjusted survival associated with lockdown a weekly time series of risk adjusted survival rate per 100 patients was plotted for the period 29 October 2018 to 16 May 2021. The weekly W statistics were calculated for each consecutive weekly period using the conventional TARN method [15,16]. The W can be interpreted as the number of excess survivors per 100 patients (observed–expected given case mix) or percentage of survivors greater or less than that expected. ITS analysis was conducted to assess the impact of the lockdowns on the baseline trend of risk-adjusted survival. A segmented regression model predicting the weekly risk-adjusted survival was estimated, and a discontinuity in the gradient (trend) or intercept (level) of the fitted model was tested for at the weekly time point of implementation of each lockdown (24 March 2020 and 2 November 2020) and at the time of relaxation of the first lockdown (29 June). The Prais–Winsten transformation was

used to adjust for auto-correlation [17]. Dates were chosen to incorporate the week each policy was implemented.

### Ethics

TARN has approval from the Health Research Authority Clinical Advisory Group (CAG–PIAG Section 251) for analysis of anonymised data.

## Results

### Population selection

Fig 1 shows the identification of the study cohort and subgroups across the time period (January 2017 to September 2021) of the time series analysis including the comparative analysis during the 2 lockdowns (Period 1: 24 March 2020 to 3 July 2020; Period 2: 1 November 2020 to 16 May 2022) and equivalent prelockdown periods.

### Changes in number of trauma presentations

Fig 2 presents the time series for number of trauma presentations for all trauma-receiving hospitals in England and MTCs from the first quarter of 2017 to the third quarter of 2021. During this period, there were 235,171 major trauma patients, and they were used to assess changes in mechanism of injury over time. For the first 36 months, major trauma patients increased in number from 16,815 in the first quarter of 2017 to 20,786 in the third quarter of 2019, with a seasonal pattern of a fall in the first quarter of each year followed by third quarter incident peaks. However, there was an unusual, sustained fall in trauma presentations from the fourth quarter of 2019 to the second quarter of 2020 to reach an all-period low of 15,986 patients, coincident with the time the first lockdown was implemented. In the third quarter of 2020, presentations rapidly increased and returned to pre-COVID levels. The second lockdown was not immediately associated with a reduction in patient numbers (November and December 2020) until the first quarter of 2021 (Fig 2). Over successive lockdowns, the pattern of variation in

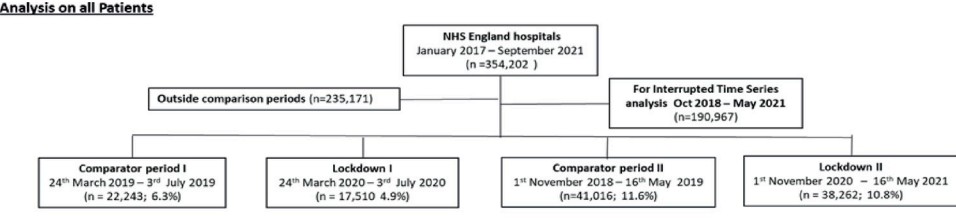

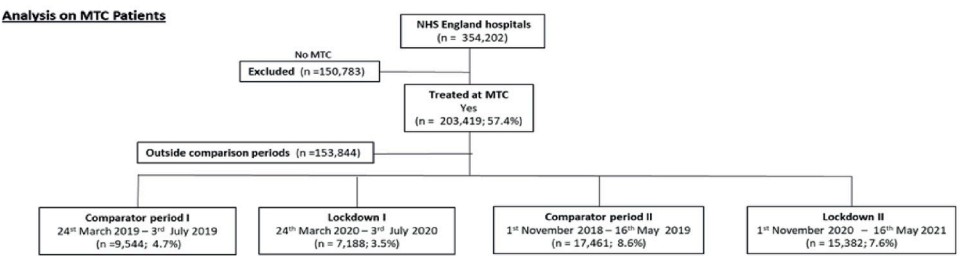

**Fig 1. Strobe diagram for inclusion of study population.**

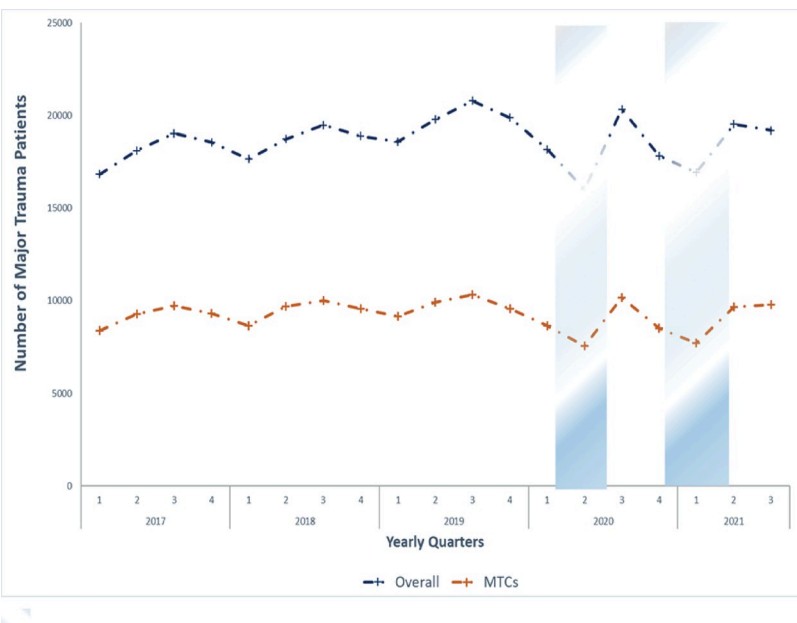

Periods of shading indicate lock-down restrictions; Major Trauma Centre (MTC)

**Fig 2. Number of major trauma patients over time.** Periods of shading indicate lockdown restrictions. MTC, major trauma centre.

volume of major trauma was similar between MTCs and all trauma-receiving hospitals—however, we observed some variation by major trauma network (S1 Fig). Fig 3 shows overall changes in the number of trauma presentations stratified by the most common mechanisms of injury. There was a large decrease in both road traffic and low falls (<2 metres) related trauma associated with the period of first lockdown restrictions; however, in the period of the second lockdown, trauma related to low falls increased. Low falls accounted for the largest number of trauma presentations and increased from the second quarter of 2018 (10,791) to peak in the fourth quarter of 2019 (12,895). Presentations then reduced over the first and second quarter of 2020 to reach a nadir (10,281). The number of low fall–related presentations then returned to a similar level prior to the lockdown, with a small decrease associated with the reintroduction of lockdown measures in the fourth quarter of 2020; however, numbers increased over the period of restrictions, reaching 12,544 in the second quarter of 2021. There were less obvious changes in injuries related to sports, high falls, or violent intent over the lockdown periods (Fig 3).

Fig 4 presents the variation in road traffic–related trauma over the study period with a comparison to measured road traffic from the Department for Transport. Before the lockdown periods, there was as underlying seasonal change where both car occupant and pedestrian trauma peaked in the fourth quarter of each year (Fig 4A). During the COVID period, there was a sustained decrease from the fourth quarter of 2019 (car occupants: 1,163; pedestrians, 840) to an all-study period low in the second quarter of 2020 (car occupants: 500; pedestrians, 225). This corresponded to an over 50%, statistically significant, reduction in trauma in these categories compared to the equivalent pre-COVID period (Table 1). Department for Transport estimates found a decrease in traffic, as measured using an index of all motor traffic, from a peak in the first quarter of 2020 of over 130, to reach a study period low in the second quarter of 2021 of around 100 (Fig 4B). Following an increase as lockdown measures were relaxed, car occupant and pedestrian trauma then fell again to a second nadir as lockdown measures were

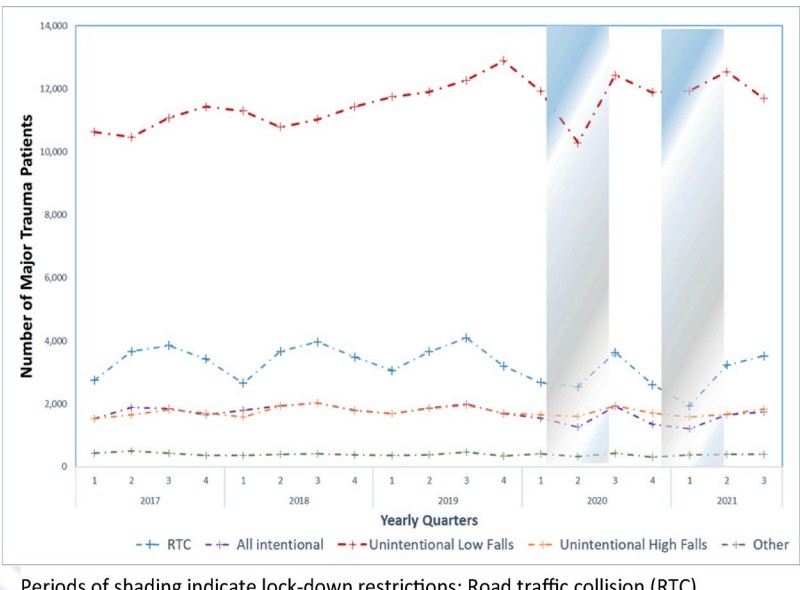

Periods of shading indicate lock-down restrictions; Road traffic collision (RTC)

**Fig 3. Number of major trauma patients presenting over time.** Periods of shading indicate lockdown restrictions. RTC, road traffic collision.

reintroduced in the first quarter of 2021 (car occupants: 602; pedestrians, 363). However, cycling-related trauma followed a different pattern, increasing over the period of the first lockdown with a peak in the third quarter of 2020 (third quarter 2020: 1,153), before returning to numbers similar to the pre-COVID period.

Fig 5 presents variation in the number of trauma presentations related to intentional injury. Blunt assault (with or without weapons) accounted for the highest number of intentional injuries with peaks in the third quarter of 2018 (1,305) and 2019 (1,258), before being followed by a sustained fall that reached a nadir in the second quarter of 2020 (670). As lockdown measures were eased numbers return to near prelockdown levels (1,137) in the third quarter of 2020 before reaching an all-study period low of 587 in the first quarter of 2021 as measures were reintroduced. Major trauma related to stabbings followed a similar pattern with study-period lows in the second (282) and fourth (269) quarter of 2020 and interspersed with peak (397) in the third quarter of 2020. Traumatic self-harm increased from a low in the fourth quarter 2019 (225) through the period of the first lockdown to peak in the third quarter of 2020 (312) before returning to pattern similar to the pre-COVID period. The small numbers of major trauma patients injured by shootings and childhood (<16 years) nonaccidental injury make changes associated with the lockdown more difficult to identify, but there is no clear pattern.

## Comparison of lockdown and equivalent pre-COVID periods

Table 1 compares characteristics of major trauma patients during the period of the first lockdown (24 March 2020 to 3 July 2020) and second lockdown (1 November 2020 to 16 May 2021) to equivalent pre-COVID periods in 2019 and 2018/2019. In total, 119,031 major trauma patients presented either during lockdown or equivalent pre-COVID periods. There was a larger decrease in the total number of trauma presentations associated with the first lockdown (absolute change −4,733 (21%)) than the second lockdown (−2,754 (6.7%)) compared to pre-COVID periods. With the largest reductions occurring for those aged between 1 and 15

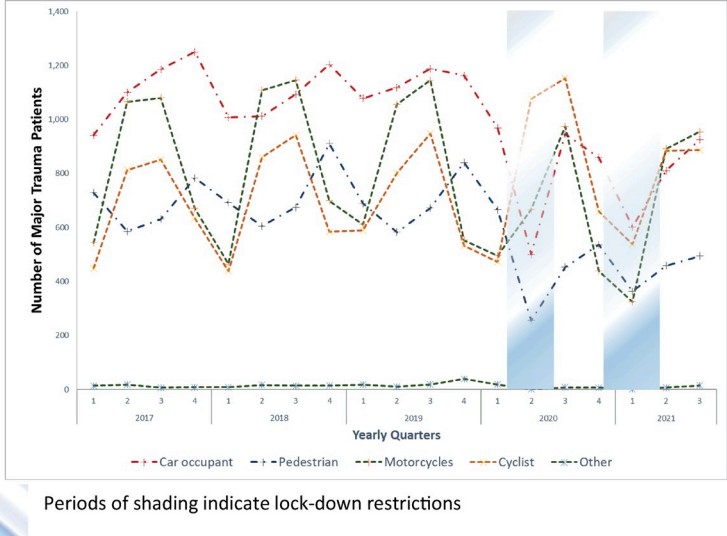

Periods of shading indicate lock-down restrictions

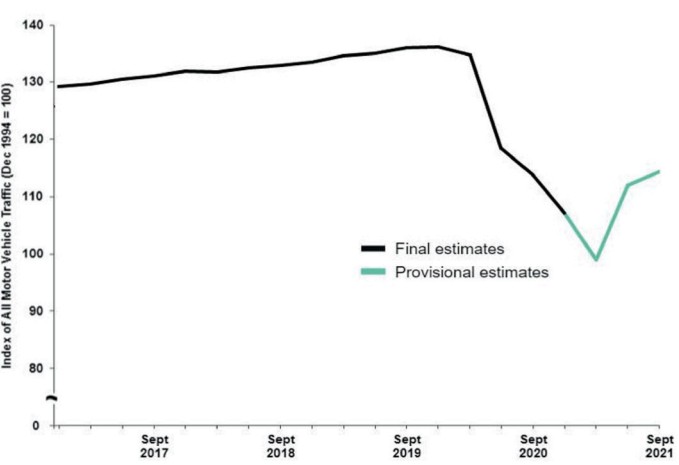

**Fig 4. Number of major trauma patients due to road traffic–related trauma compared to index of moto vehicular traffic in England over time.** (**a**) Number of major trauma patients presenting due to road traffic injuries over time. (**b**) Rolling annual indices of road traffic in Great Britain (reproduced from Department of Transport website with permission). Periods of shading indicate lockdown restrictions.

and 16 and 64 for both lockdowns. The age groups of 65 and over and 85 and over experienced smaller reductions in trauma presentations in the first lockdown (−1,878 (16%) and −563 (12.2%)) and then increased during the second lockdown (65 and over: +665 (3%); 85 and over +828 (9.3%)). This was accompanied by a reduction in the proportion of patients aged 16 to 64 in both lockdown periods compared to the comparator periods (lockdown period 1: −3.2% (95% CI −4.1% to −2.2%); lockdown period 2: −5.3% (−6% to −5%)) and increases in the proportion aged 65 and over (lockdown period 1: 3.5% (95% CI 2.5% to 4.5%); lockdown period 2: 5.7% (5% to 6.3%)) and 85 and over (lockdown period 1: 2.4% (95% CI 1.6% to 3.2%); lockdown period 2: 3.7% (3.1% to 4.3%)) (Table 1). This corresponded to a 4-year increase in the median age of trauma patients and number of patients with a moderate and high Charlson comorbidity index (between 6 to 10 and >10 during both lockdown periods).

**Table 1. Comparison of demographics pre-COVID and lockdown periods.**

| | Period | | | Period | | |
|---|---|---|---|---|---|---|
| | 24 March 2019–03 July 2019 (comparator) N (%within cohort) | 24 March 2020–03 July 2020 (lockdown 1) N (%within cohort) | Absolute change [percentage change proportion (95%CI)] | 01 November 2018–16 May 2019 (comparator) N (%within cohort) | 01 November 2020–16 May 2021 (lockdown 2) N (%within cohort) | Absolute change [percentage change proportion (95%CI)] |
| Total | 22,243 | 17,510 | −4,733 (−21%) | 41,016 | 38,262 | 2,754 (−6.7%) |
| Age (years), Median (IQR) | 67.6 (46.5–83.1) | 70.9 (50.3–84.2) | 3.3 (4.5%) | 69.1 (48.7–83.6) | 73.1 (53.3–85.1) | 4 (5.8%) |
| **Age bands, n(%)** | | | | | | |
| Age <1 | 138 (0.6%) | 130 (0.7%) | −8 (−6.1%) [0.1(−0.04, 0.030)] | 281 (0.7%) | 234 (0.6%) | −47 (−16.7%] [0.1 (−0.2, 0.04)] |
| Age <16 | 942 (4.2%) | 674 (3.8%) | −268 (−28.4%) [−0.4 (−0.8, 0)] | 1,444 (3.5%) | 1,218 (3.2%) | −226 (−15.6%) [0.3(−0.6, − 0.1)] |
| Age 16–64 | 9,561 (43%) | 6,974 (39.8%) | −2,587 (−27.5%) [−3.2(−4.1, −2.2)] | 17,173 (41.9%) | 13,980 (36.5%) | −3,193 (−18.6%) [−5.3(−6, −5)] |
| Age 65 and over | 11,740 (52.8%) | 9,862 (56.3%) | −1,878 (−16%) [3.5 (2.5, 4.5)] | 22,399 (54.6%) | 23,064 (60.3%) | 665 (3%) [5.7(5, 6.3)] |
| Age 85 and over | 4,610 (20.7%) | 4,047 (23.1%) | −563 (−12.2%) [2.4 (1.6, 3.2)] | 8,903 (21.7%) | 9,731 (25.4%) | 828 (9.3%) [3.7 (3.1, 4.3)] |
| Male, n (%) | 12,316 (55.4%) | 9,512 (54.3%) | −2,804 (−22.8%) [−1 (−2, −0.6)] | 22,146 (54%) | 19,769 (51.7%) | −2,377 (−10.7%) [−2.3 (−3, −1.6)] |
| **CCI*, n(%)** | | | | | | |
| CCI 0 | 9,359 (42.1%) | 6,220 (35.5%) | −3,139 (−33.5%) [−6.5 (−7.5, −5.6)] | 16,665 (40.6%) | 12,806 (33.5%) | −3,859 (−23.2%) [−7.1(−7.8, −6.5)] |
| CCI 1–5 | 8,538 (38.4%) | 6,896 (39.4%) | −1,642 (−19.2%) [1 (0.3, 2)] | 15,899 (38.8%) | 15,667 (40.9%) | −232 (−1.5%) [2.2 (1.5, 2.9)] |
| CCI 6–10 | 3,032 (13.6%) | 3,061 (17.5%) | 29 (0.96%) [3.8 (3.2, 4.6)] | 5,987 (14.6%) | 6,863 (17.9%) | 876 (14.6%) [3.3(2.8, 3.8)] |
| CCI >10 | 927 (4.2%) | 1,024 (5.8%) | 97 (10.5%) [1.7(1.2, 2.1)] | 1,648 (4%) | 2,410 (6.3%) | 762 (46.2%) [2.3(2, 2.6)] |
| Not recorded | 387 (1.7%) | 309 (1.8%) | −88 (−22.7%) [0.2 (−0.2, 0.3)] | 817 (2%) | 516 (1.3%) | −301 (36.8%) [−0.6(−0.8, −0.5)] |
| **MOI§: RTC, n(%)** | | | | | | |
| Car occupant | 1,247 (30.7%) | 551 (20.4%) | −696 (−55.8%) [−10.4(−12.4, −8.2)] | 2,485 (35.2%) | 1,551 (31.3%) | −934 (−37.6%] [−3.9(−5.6, −2.2)] |
| Pedestrian | 661 (16.3%) | 288 (10.6%) | −373 (−56.4%) [−5.6 (−7.2, −4)] | 1,629 (23.1%) | 962 (19.4%) | −667 (−40.9%) [−3.7(−5.1, −2.2)] |
| Motorcycles | 1,196 (29.4%) | 711 (26.3%) | −485 (−40.5%) [−3.2(−5.3, −1)] | 1,524 (21.6%) | 976 (19.7%) | −548 (−35.9%) [−1.9(−3.3, −0.4)] |
| Cyclist | 912 (22.4%) | 1,139 (42.1%) | 227 (24.9%) [19.6 (17.4, 21.9)] | 1,315 (18.6%) | 1,396 (28.2%) | 81 (6.2%) [9.5(8, 11.1)] |
| Other | 11 (0.3%) | 1 (0%) | −10 (−90%) [−0.2(−0.4, −0.06)] | 31 (0.4%) | 10 (0.2%) | −21 (−67.7%) [−0.23(−0.4, −0.04)] |
| **MOI: Intentional, n(%)** | | | | | | |
| Blunt assault | 130 (0.6%) | 88 (0.5%) | −42 (−32.3%) [−0.08 (−0.2, 0.06)] | 227 (0.6%) | 175 (0.5%) | −52 (−22.9%) [−0.1(−0.2, 0.002)] |
| Self-harm | 276 (1.2%) | 284 (1.6%) | 8 [2.9%] [0.4 (0.1, 0.6)] | 525 (1.3%) | 562 (1.5%) | 37 [7%] [0.2 (0.02, 0.3)] |
| NAI | 63 (0.3%) | 27 (0.2%) | −36 (−57.1%) [−0.1(−0.2, −0.03)] | 97 (0.2%) | 90 (0.2%) | −7 (−7.2%) [−0.001(−0.07 to 0.07)] |
| Shooting | 34 (0.2%) | 40 (0.2%) | 6 (17.6%) [0.08(−0.01, 0.2)] | 80 (0.2%) | 56 (0.1%) | −24 (−30%) [−0.05(−0.1, 0.001)] |

(*Continued*)

**Table 1.** (Continued)

| | Period | | | Period | | |
|---|---|---|---|---|---|---|
| | 24 March 2019–03 July 2019 (comparator) N (%within cohort) | 24 March 2020–03 July 2020 (lockdown 1) N (%within cohort) | Absolute change [percentage change proportion (95%CI)] | 01 November 2018–16 May 2019 (comparator) N (%within cohort) | 01 November 2020–16 May 2021 (lockdown 2) N (%within cohort) | Absolute change [percentage change proportion (95%CI)] |
| Stabbing | 450 (2%) | 312 (1.8%) | −138 (−30.7%) [−0.2(−0.5, 0.03)] | 791 (1.9%) | 589 (1.5%) | −202 (−25.5%) [−0.4 (−0.6, −0.2)] |
| Blows | 1,174 (5.3%) | 647 (3.7%) | −527 (−44.9%) [−1.6(−1.9, −1.2)] | 2,059 (5%) | 1,299 (3.4%) | −760 (−36.9%) [−1.6(−1.9, −1.3)] |
| **Unintentional, n (%)** | | | | | | |
| Falls >2 m | 2,055 (9.2%) | 1,757 (10%) | −298 (−14.5%) [0.8(0.2, 1.4)] | 3740 (9.1%) | 3,528 (9.2%) | −212 (−5.7%) [0.1(−0.3, 0.5)] |
| Falls <2 m | 13,384 (60.2%) | 11,314 (64.6%) | −2,070 (−15.5%) [4.4 (3.5, 5.4)] | 25,505 (62.2%) | 26,203 (65.8%) | 698 (2.7%) [6.3 (5.6, 6.9)] |
| Sport | 449 (2%) | 320 (1.8%) | −129 (−28.7%) [−0.2 (−0.5, 0.01)] | 615 (1.5%) | 489 (1.3%) | −126 (−20.5%) [−0.2 (−0.4, −0.006)] |
| **GCS bands, n(%)** | | | | | | |
| Mild | 19,609 (88.2%) | 15,449 (88.2%) | 4,160 (21.2%) [0.1 (−0.6, 0.7)] | 35,831 (87.4%) | 34,051 (89%) | −1,780 (−5%) [1.6 (1.2, 2.1)] |
| Moderate | 689 (3.1%) | 625 (3.6%) | −64 (−9.3%) [0.5(0.1, 0.8)] | 1,333 (3.2%) | 1,127 (2.9%) | −206 (−15.4%) [−0.3 (−0.5, −0.06)] |
| Severe | 955 (4.3%) | 765 (4.4%) | −190 (−19.9%) [0.1 (−0.3, 0.5)] | 1,886 (4.6%) | 1,464 (3.8%) | −422 (−22.4%) [−0.8(−1, −0.5)] |
| Not recorded | 990 (4.5%) | 671 (3.8%) | −319 (−32.2%) [−0.6(−1, −0.2)] | 1,966 (4.8%) | 1,620 (4.2%) | −346 (−17.6%) [−0.6(−0.8, −0.3)] |
| ISS[***], median (IQR) | 9 (9–18) | 9 (9–18) | 0 | 9 (9–18) | 9 (9–17) | 0 |
| **ISS bands, n(%)** | | | | | | |
| ISS 1–8 | 4,545 (20.4%) | 3,062 (17.5%) | −1,483 (−32.6%) [−3 (−4, −2)] | 8,266 (20.2%) | 7,838 (20.5%) | −428 (−5.2%) 0.3(−0.2, 0.9)] |
| ISS 9–15 | 9,290 (41.8%) | 7,728 (44.1%) | −1,562 (−16.8%) [2.4(1.4, 3.3)] | 17,207 (42%) | 16,969 (44.3%) | −233 (−1.4%) [2.4(1.7, 3.1)] |
| ISS >15 | 8,408 (37.8%) | 6,720 (38.4%) | −1,688 (−20.1%) [5.6(−0.4, 1.5)] | 15,543 (37.9%) | 13,455 (35.2%) | −2,088 (−13.4%) [−2.7 (−3.4,-2)] |
| ISS >25 | 3,995 (18%) | 3,127 (17.9%) | −868 (−21.7%) [−0.1(−0.9, 0.7)] | 7,521 (18.3%) | 6,201 (16.2%) | −1,320 (−17.6%) [−2.1(−2.6, −1.6)] |
| **Body regions, n (%)** | | | | | | |
| Head AIS 3+ | 5,911 (26.6%) | 4,670 (26.7%) | −1,241 (−21%) [0.1 (−0.8, 1)] | 11,128 (27.1%) | 9,629 (25.2%) | −1,499 (−13.5%) [−2(−2.6 to −1.3)] |
| Face AIS 3+ | 63 (0.3%) | 41 (0.2%) | −22 (−34.9%) [−0.05 (−0.1, 0.05)] | 99 (0.2%) | 69 (0.2%) | −30 (−30.3%] [−0.06 (−0.1 to 0)] |
| Chest AIS 3+ | 4,787 (21.5%) | 3,915 (22.4%) | −872 (−18.2%) [8.3 (0.2, 1.6)] | 8,515 (20.8%) | 8,075 (21.1%) | −440 (−5.2%] [0.3 (−0.2 to 0.9)] |
| Abdomen AIS 3+ | 872 (3.9%) | 690 (3.9%) | −182 (−20.9%) [0.02 (−0.3, 0.4)] | 1,465 (3.6%) | 1,179 (3.1%) | −286 (−19.5%) [−0.5 (−0.7, −0.2)] |
| Spine AIS 3+ | 1,985 (8.9%) | 1,561 (8.9%) | −424 (−21.4%) [−0.01(−0.6, 0.5)] | 3,784 (9.2%) | 3,459 (9%) | −325 (−8.6%] [−0.2(−0.6, 0.2)] |
| Pelvis AIS 3+ | 758 (3.4%) | 600 (3.4%) | −158 (−20.8%) [0.02(−0.3, 0.4)] | 1,501 (3.7%) | 1,386 (3.6%) | −115 (−7.7%) [−0.04(−0.3, 0.2)] |
| Limb AIS 3+ | 5,707 (25.7%) | 4,892 (27.9%) | −815 (−14.3%) [2.3 (1.4, 3.2)] | 10,719 (26.1%) | 10,122 (26.5%) | −597 (−5.6%) [0.3(−0.3, 0.9)] |

(*Continued*)

**Table 1.** (Continued)

| | Period | | | Period | | |
|---|---|---|---|---|---|---|
| | 24 March 2019–03 July 2019 (comparator) N (%within cohort) | 24 March 2020–03 July 2020 (lockdown 1) N (%within cohort) | Absolute change [percentage change proportion (95%CI)] | 01 November 2018–16 May 2019 (comparator) N (%within cohort) | 01 November 2020–16 May 2021 (lockdown 2) N (%within cohort) | Absolute change [percentage change proportion (95%CI)] |
| Other AIS 3+ | 217 (1%) | 199 (1.1%) | −18 (−8.3%) [0.2 (−0.04, 0.3)] | 375 (0.9%) | 396 (1%) | 21 (5.6%) [0.1 (−0.01, 0.2)] |
| Polytrauma | 1,622 (7.3%) | 1,350 (7.7%) | −272 (−16.8%) [0.4 (−0.1, 0.9)] | 2,984 (7.3%) | 2,429 (6.3%) | −555 (−18.6%) [−0.9(−1.2, 0.6)] |

\* CCI, Charlson comorbidity index.

§ MOI, mechanism of injury.

\*\*\* ISS, Injury Severity Score.

There was a significant reduction in the proportion of patients with ISS injuries 1 to 8 (−3% 95% CI −4% to −2%) during the first lockdown and ISS injuries >15 during the second lockdown period (−2.7% 95% CI −3.4% to −2%). The largest absolute reductions in severity and type of injury were for mild (−4,160 (21.2%) and severe (−190 (19.9%)) traumatic brain injuries; ISS injuries <8 (−1,483 (32.6%)) and facial injuries (−22 (34.9%)) during the first lockdown. During the second lockdown period, the largest absolute reductions were for moderate (−206 (15.4%)) and severe (−422 (22.4%)) traumatic brain injury; ISS injuries >25 (−1,320 (17.6%)), facial (−30 (30.3%)), and abdominal injuries (−286 (19.5%)).

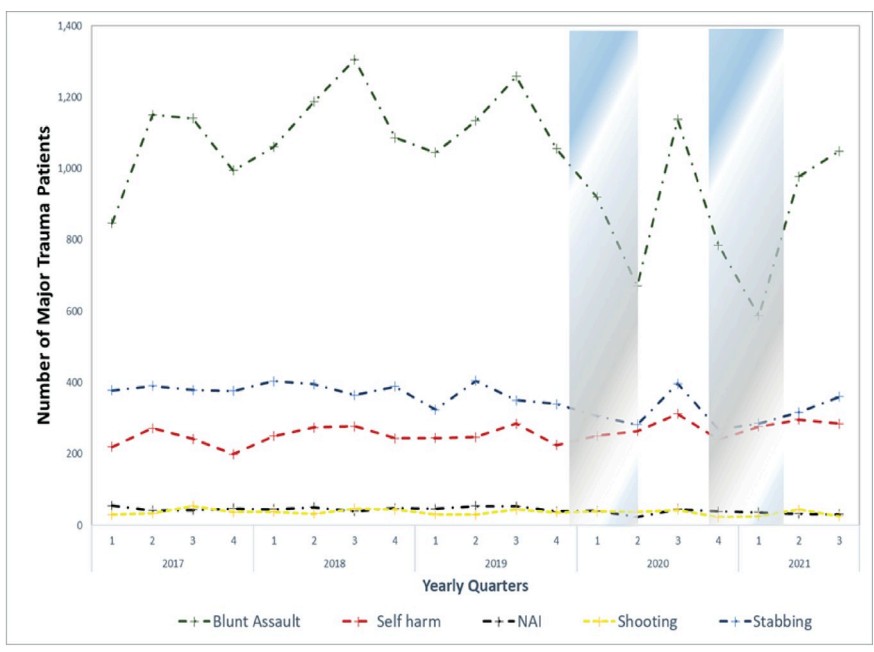

Periods of shading indicate lock-down restrictions; Non-accidental injury (NAI)

**Fig 5. Number of major trauma patients due to intentional injury over time.** Periods of shading indicate lockdown restrictions. NAI, nonaccidental injury.

Table 2 compares the treatment pathways and outcomes of major trauma patients during the period of the first lockdown (24 March 2020 to 3 July 2020) and second lockdown (1 November 2020 to 16 May 2021) to equivalent pre-COVID periods in 2019 and 2018/2019). During the first lockdown period, there were large absolute and small but statistically significant reductions in proportions of patients treated in an MTC (pre-COVID: 11,176 (50.2%); lockdown 1: 8,256 (47.2%)), seen by a consultant in the ED (pre-COVID: 8,140 (36.6%); lockdown 1: 5,562 (31.8%)) and treated in the intensive care unit (ICU) (pre-COVID: 3,092 (13.9%); lockdown 1: 2,208 (12.6%)). These reductions were also observed in the second lockdown period.

## Interrupted time series analysis

A small increase in the proportion of patients who died was observed in both lockdown periods compared to the pre-COVID periods (Table 2). Fig 6 shows the weekly time series for adjusted likelihood for survival (W-Score) from 29 October 2018 to 16 May 2021; this analysis included 190,967 major trauma patients. Before the first lockdown, there was a nonstatistically significant tendency for reducing survival (−0.007; 95% CI:-0.016 to 0.001). The introduction of the first lockdown in the week of 23 March 2020 was associated with a statistically significant reduction in the level of likelihood of survival (−1.71; 95% CI: −2.76 to −0.66) and statistically significant reversal trend (0.25; 95% CI: 0.14 to 0.35) (Fig 6). This equated to around 1 excess death per 100 patients for first 6 weeks of the lockdown restrictions (Fig 6).

Easing of restrictions in the week of 29 June was associated with a statistically significant reversal in trend (−0.32 95% CI: −0.45 to −0.19) and likelihood of survival is similar to the period immediately before lockdown before COVID restrictions are reintroduced in the week of 2 November 2021. This reimplementation of lockdown measures is associated with a smaller and nonstatistically significant immediate reduction in level and chance of survival (−0.62 95% CI: −1.74 to 0.50) and further statically significant reversal in trend (0.14 95% CI: 0.06 to 0.22).

## Discussion

### Summary

We have conducted a national cohort study using trauma registry data to assess the impact of successive lockdowns on trauma presentations and treatment pathways in England. The first lockdown had a larger associated reduction in total trauma volume (−21%) compared to the pre-COVID period than the second lockdown (−6.7%). There were large reductions in trauma related to car occupant and pedestrian road traffic accidents associated with both lockdowns but an increase in cyclist-related trauma, particularly during the first lockdown (Fig 4A and Table 1). Both lockdowns were associated with an increase in the average age of trauma patients compared to the pre-COVID comparator periods (Table 1). Smaller reductions in trauma were observed for those 85 and over (−12.2%) with the first lockdown and trauma volume increased for those 65 and over (3%) and 85 and over (9.3%) during the second lockdown, with corresponding increases in comorbidity (Table 1).

During both lockdown periods, a smaller proportion of patients were bypassed or transferred to MTCs, were received by a consultant, and admitted to ICU (Table 2). A small increase in absolute mortality was also observed compared to the pre-COVID period. To specifically assess whether changes in mortality were due to changes in case mix or clinical care, we conducted an ITS analysis for adjusted likelihood for survival. We found a reduction in level of likelihood of survival (−1.71; 95% CI: −2.76 to −0.66) associated with the introduction of the first lockdown, indicating that either changes in care pathways due to the pandemic, or patient characteristics not adjusted for, may have initially impacted trauma care and related

**Table 2. Comparison of care pathways pre-COVID and lockdown periods.**

| | 24 March 2019–03 July 2019 (comparator) N (%within cohort) | 24 March 2020–03 July 2020 (lockdown 1) N (%within cohort) | Absolute Change [percentage change] p-value‡ | 01 November 2018–16 May 2019 (comparator) N (%within cohort) | 01 November2020–16 May 2021 (lockdown 2) N (%within cohort) | Absolute Change [percentage change] p-value‡ |
|---|---|---|---|---|---|---|
| First Hospital MTC | 9,908 (44.5%) | 7,376 (42.1%) | −2,532 (−25.6%) −2,532 [−2.4 (−3.4, −1.4)] | 18,099 (44.1%) | 15,928 (41.6%) | −2,171 (−12%) [−2.5 (−3.2 to −1.8)] |
| Treated at MTC | 11,176 (50.2%) | 8,256 (47.2%) | −2,920 (−26.1%) [−3 (−4 to −2)] | 20,395 (49.7%) | 17,852 (46.7%) | −2,543 (−12.5%) [−3 (−4 to −2.4)] |
| Consultant ED | 8,140 (36.6%) | 5,562 (31.8%) | −2,578 (−31.7%) [−4.8(−5.8, −3.9)] | 14,779 (36%) | 12,577 (32.9%) | −2,202 (−14.9%) [−3.2 (−3.8, −2.5)] |
| CT within 1 hour | 5,062 (31.9%) | 3,992 (30.9%) | −1,070 (−21.1%) [−0.9(−2, 0.1)] | 9,203 (31.6%) | 7,776 (27.1%) | −1,427 (−15.5%) [−4(−5, −3.7)] |
| Whole body CT | 3,348 (15.1%) | 3,210 (18.3%) | −138 (−4.1%) [3 (2, 4)] | 6,040 (14.7%) | 6,417 (16.8%) | 377 (6.2%] p = 0.001** |
| ICU stay | 3,092 (13.9%) | 2,208 (12.6%) | −884 (−28.6%) [−1.3(−1.9, −0.6)] | 5,591 (13.6%) | 3,850 (10.1%) | −1,741 (−31.1%) [−3.6(−4, −3)] |
| Mortality* | 1,417 (7.1%) | 1,316 (8.3%) | −101 (−7.1%) [1.2 (0.6, 1.7)] | 2,916 (7.9%) | 2,858 (8.1%) | −58 (−2%) [0.2 (−0.1, 0.6)] |
| **Discharge destination, n(%)** | | | | | | |
| Home (own) | 13,800 (62%) | 10,484 (59.9%) | −3,316 (−24%) [−2(−3.1, −1.2)] | 24,961 (60.9%) | 23,368 (61.1%) | −1,593 (−6.4%) [−0.7 (−1.4, −0.05)] |
| Home (relative/ carer) | 473 (2.1%) | 372 (2.1%) | −101 (−21.4%) [0 (−0.3, 0.3)] | 974 (2.4%) | 852 (2.2%) | −122 (−12.5%) [−0.1(−0.4, 0.06)] |
| Mortuary* | 1,501 (6.7%) | 1,323 (7.6%) | −178 (−11.9%) [0.8(0.3, 1.3)] | 3,086 (7.5%) | 2,977 (7.8%) | −109 (−3.5%) [0.1 (−0.3, 0.5)] |
| No fixed abode | 75 (0.3%) | 47 (0.3%) | −28 (−37.3%) p = 0.218 | 107 (0.3%) | 87 (0.2%) | −20 (−18.7%) p = 0.340 |
| Not Known | 87 (0.4%) | 39 (0.2%) | −48 (−55.2%) p < 0.003 | 101 (0.2%) | 95 (0.2%) | −6 (−5.9%) p = 0.954 |
| Nursing Home | 1,190 (5.3%) | 1,063 (6.1%) | −127 (−10.7%) [0.7(0.3, 1.2)] | 2,448 (6%) | 2,231 (5.8%) | −217 (−8.9%) [−0.2(−0.6, 0.1)] |
| Other Acute hospital | 2,425 (10.9%) | 1,736 (9.9%) | −689 (−28.4%) [−0.1(−1.6, −0.4)] | 4,346 (10.6%) | 3,313 (8.7%) | −1,033 (−23.8%) [−0.1(−0.5, 0.2)] |
| Other institution | 526 (2.4%) | 516 (2.9%) | −10 (−1.9%] [0.6 (0.3 to 0.9)] | 980 (2.4%) | 870 (2.3%) | −110 (−11.2%) [−0.1 (−0.3, 0.1)] |
| Rehabilitation | 2,077 (9.3%) | 1,871 (10.7%) | −206 (−9.9%) [1.3(0.7, 1.9)] | 3,851 (9.4%) | 4,274 (11.2%) | 423 (11%] [1.7(1.3, 2.2)] |
| Social care | 63 (0.3%) | 50 (0.3%) | −13 (−20.6%) [0 (−0.1, 0.1)] | 121 (0.3%) | 103 (0.3%) | −18 (−14.9%) [−0.2(−0.1, 0.5)] |

* These totals do not correspond as mortality includes deaths in the community and is censored at 30 days.

** Statistical significance after Bonferroni adjustment.

ED, emergency department; ICU, intensive care unit; MTC, major trauma centre.

outcomes. This was followed by a trend of improving survival (0.25; 95% CI: 0.14 to 0.35) during the first lockdown period (Fig 6). The trend of improving survival reversed as lockdown measures were relaxed and then reversed again to a trend of improving as restrictions were reintroduced during the second lockdown (Fig 6).

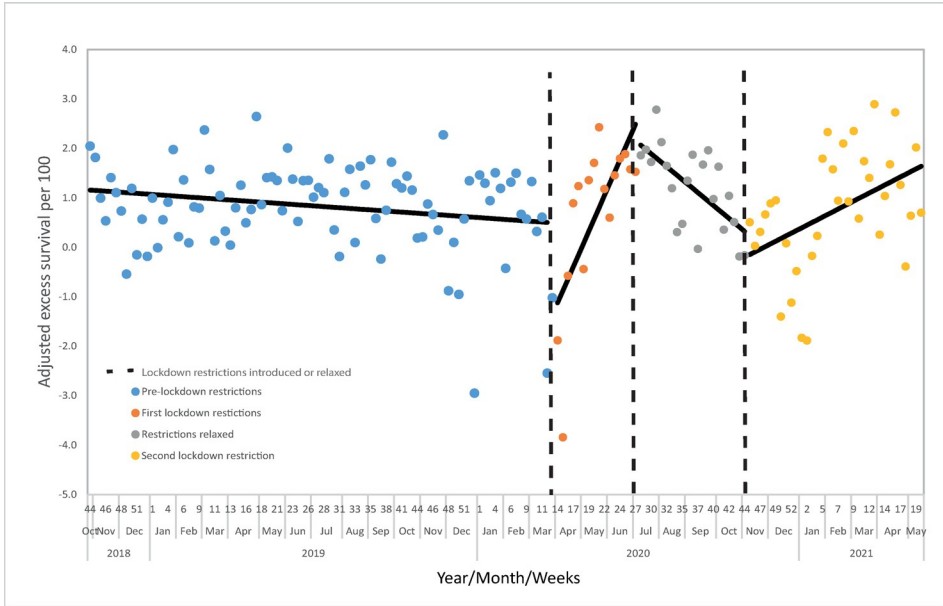

**Fig 6. Interrupted time series analysis assessing the impact of COVID restrictions on likelihood of survival following major trauma.** Segmented regression model (fitted solid lines) predicting the weekly risk adjusted survival with discontinuities tested for at implementation of each lockdown (24 March 2020 and 2 November 2020) and at the time of relaxation of the first lockdown (29 June).

## Comparison to previous literature

This is one of the few national evaluations of the impact of COVID-related measures on trauma care and outcomes, with previous UK studies limited to single-centres and descriptive analysis [12,18–20]. One study found no change in mortality during the first lockdown compared to an equivalent pre-COVID period in patients with ISS >15 injuries in London [18]. Another study found that although the mortality was higher in the first lockdown period at an English MTC, following adjustment for age, injury severity, and frailty, there was no increased risk of death associated with the lockdown period. In a recent review of international epidemiology of major trauma of 35 included studies, 26 were single-centre and 9 were multicentre studies (either hospital network or region) with before and after designs [8]. Our use of a complete national dataset and a quasi-experimental method provides more robust evidence of the impact of lockdown measure of trauma outcomes in England, identifying that the first lockdown may have been associated with an initial reduction in likelihood of survival, which improved as the pandemic progressed.

The only European study we could identify that used an equivalent national trauma registry found a reduction in ICU admission for traumatically injured patient in the Netherlands was associated with higher than predicted trauma-related mortality rate and risk of death in Traumatic Brain Injury (TBI) patients not admitted to ICU during the first wave of the pandemic compared to a pre-COVID comparator period [7]. A meta-analysis of before and after studies from 14 countries found no evidence of increased trauma-related mortality during the first wave of the pandemic [8]. Our method, robustly adjusting both for changes case mix and trend over time, found a similar signal of reduced survival following trauma at the beginning of the lockdown period. However, the subsequent trend indicates that this may not simply be due to availability of ICU care and may relate to staff having to adjust to different working environments and new care pathways. In particular, the management of low energy trauma

(low falls) in older adults, of which there was a higher proportion during the lockdown periods, does not generally require care in ICU but may require multidisciplinary care and input from multiple clinical teams that may have been disrupted by reconfiguration of service for the pandemic.

In terms of changes to mechanism of injury, our results are broadly similar to those described in a previous review, with the largest reduction in trauma being observed for road traffic accidents and small increases in self-harm over the period COVID restrictions are introduced [8,20]. The granularity of our data allowed us to identify an increase in cycling-related trauma during the first lockdown period in England, which appears to be a novel finding. The evidence of the impact of COVID restrictions on interpersonal violence is more mixed. Our study and other UK studies have found reductions in interpersonal violence during lockdown periods [19,21]; however, studies conducted in the United States of America, France, and India found interpersonal violence and, particularly, domestic violence to have increased during restrictions, as did the findings of a large review [8,21].

## Strengths and limitations

We have used a large national trauma (TARN) registry for our analysis, which has robust mechanisms to ensure completeness and quality of data collection at all trauma-receiving hospital in England. The TARN inclusion criteria means that injured patients who die in the pre-hospital setting, or patients who are not admitted for >72 hours, nor admitted to critical care, nor transferred to a tertiary/specialist centre, nor die within 30 days of injury, are excluded from our study population. Only 4% of deaths due to suicide occur within hospital [22], which may mean that we have underestimated the burden of the most serious injuries related to self-harm associated with the lockdown period. Misclassification of injuries related to domestic violence as other types of interpersonal violence, along with the lower severity of many of these injuries, means that our study may not have been sensitive enough to identify the increases in domestic violence associated with COVID restrictions observed elsewhere [6]. Only a small number of cases of paediatric nonaccidental injuries meet the TARN inclusion criteria, and low numbers make it difficult to make any conclusions about the impact of lockdown measures.

Our use of ITS analysis adjusts both for changes in case mix and trends over time when assessing the impact of lockdown measures on likelihood of survival. However, adjustment for likelihood of survival following major trauma did not include COVID status of patients or hospital acquired infection, which may have affected likelihood of survival.

## Implications

Out study has found those aged over 65 had smaller than average reductions in trauma during the first lockdown period and increases during the second lockdown period, compared to pre-COVID comparator periods. Trauma in older adults is likely to result from low falls in the home, which may have been less affected by lockdown measures, and research is required to determine the optimal pathways for the triage and management of significant injury resulting from low energy mechanisms in older adults [23]. Additionally, our study identified that lockdown measures were associated with trends of increased likelihood of survival, while relaxation of restrictions following the first lockdown was associated with a trend of reducing survival (Fig 6). There may be due to a reducing prevalence of COVID during lockdown periods; however, COVID status was not available in our data. Different COVID restrictions with varying levels of compliance have been introduced internationally during the pandemic. A comparison of the effects of different lockdown measures' impact on trauma outcomes

internationally may be helpful in identifying which aspects of restrictions caused the changes observed in our study.

The most important finding for clinical practice is the reduction in adjusted survival identified in the ITS analysis at the beginning of the first lockdown (Fig 6). This equated initially to around 2 excess deaths per 100 patients. Although, infection with COVID was not adjusted for, such an immediate change is unlikely to be caused by changes in COVID prevalence caused by restrictions. It is likely this immediate change in likelihood of survival is related to reconfiguration and disruption of services. Despite exclusion of patients aged over 65 with isolated femoral neck or single pubic ramus fractures from the TARN registry, older adults presenting with low falls represent the majority of TARN-eligible patients with TBI the commonest cause of death in this group. Care of older adults with major trauma involves integration of multiple services, and this may have been disrupted during periods of COVID-related restrictions. Reassuringly, over the duration of the first period of COVID restrictions, adjusted likelihood of survival increased and reached prepandemic levels. The initial reduction in likelihood of survival cannot solely be attributed to increased ICU occupancy impacting on care as reduced care as the proportion of patients admitted to ICU decreased during the second lockdown period without a corresponding significant decrease in survival ICU occupancy related to COVID increased as the pandemic progressed. If similar restrictions are introduced again, measures may be needed to mitigate the observed initial negative impact on trauma outcomes.

COVID-related restrictions were associated with significant reductions in interpersonal violence, occupant, and pedestrian road traffic mechanisms for major trauma. Measures that similarly reduce road traffic volume (such as encouraging working from home) or restrict opportunities for interpersonal violence may be effective public health interventions for reducing trauma-related morbidity. The relationship between traffic volume and incidents of road traffic–related trauma, however, is not a simple linear relationship [24]. Despite sustained reductions in traffic volume observed through the period of relaxation of restrictions between the lockdowns, an increase in road traffic collisions was observed (Fig 4). Total road traffic accidents may be related to lockdown effects on the types of road vehicle (HGV numbers maintained), types of road user (fewer older drivers), and driver behaviour (speed or risk) as well as simple traffic volume. The large increase in cyclist-related accidents observed during the first period of restrictions may mean that, in summer months at least, efforts to reduce road traffic may have caused an unintended increase in cyclist-related trauma.

## Conclusions

In the first evaluation of the impact of COVID restrictions on major trauma presentations, care pathways, and outcomes in England, large reductions in overall trauma volume were observed particularly in interpersonal violence and both occupant- and pedestrian-related road traffic collisions. Future research is needed to better understand the initial reduction in likelihood of survival after major trauma observed with the implementation of the first lockdown to prevent this occurring if similar measures are introduced again.

## Supporting information

**S1 Appendix. Study protocol.**
(DOCX)

**S1 Checklist. STROBE statement.**
(DOCX)

**S1 Fig. Number of major trauma patients stratified by network over time.** Periods of shading indicate lock-down restrictions. Nwk, network.
(DOCX)

## Author Contributions

**Conceptualization:** Carl Marincowitz, Omar Bouamra, Tim Coats, Dhushy Kumar, David Lockey, Lyndon Mason, Virginia Newcombe, Julian Thompson, Antoinette Edwards, Fiona Lecky.

**Data curation:** Omar Bouamra, Antoinette Edwards.

**Formal analysis:** Carl Marincowitz, Fiona Lecky.

**Investigation:** Carl Marincowitz, Omar Bouamra.

**Methodology:** Carl Marincowitz, Omar Bouamra, Fiona Lecky.

**Project administration:** Carl Marincowitz.

**Supervision:** Tim Coats, Dhushy Kumar, David Lockey, Lyndon Mason, Fiona Lecky.

**Writing – original draft:** Carl Marincowitz, Omar Bouamra, Fiona Lecky.

**Writing – review & editing:** Carl Marincowitz, Tim Coats, Dhushy Kumar, David Lockey, Lyndon Mason, Virginia Newcombe, Julian Thompson, Fiona Lecky.

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
