## [Editor Report · Decision Letter 0]

27 Oct 2022

Dear Dr Marincowitz, 

Thank you for submitting your manuscript entitled "The effect of the COVID-19 pandemic on major trauma presentations and patient outcomes in English hospitals." for consideration by PLOS Medicine.

Your manuscript has now been evaluated by the PLOS Medicine editorial staff as well as by an academic editor with relevant expertise and I am writing to let you know that we would like to send your submission out for external peer review.

Please re-submit your manuscript within two working days, i.e. by Oct 31 2022 11:59PM.

Kind regards,

Callam Davidson

Senior Editor

PLOS Medicine

---

## [Decision Letter · Decision Letter 1]

14 Mar 2023

Dear Dr. Marincowitz,

Thank you very much for submitting your manuscript "The effect of the COVID-19 pandemic on major trauma presentations and patient outcomes in English hospitals." (PMEDICINE-D-22-03493R1) for consideration at PLOS Medicine. 

[LINK]

In light of these reviews, I am afraid that we will not be able to accept the manuscript for publication in the journal in its current form, but we would like to consider a revised version that addresses the reviewers' and editors' comments. Obviously we cannot make any decision about publication until we have seen the revised manuscript and your response, and we plan to seek re-review by one or more of the reviewers. 

We hope to receive your revised manuscript by Apr 04 2023 11:59PM. Please email us (plosmedicine@plos.org) if you have any questions or concerns.

We look forward to receiving your revised manuscript. 

Sincerely,

Callam Davidson, 

PLOS Medicine

plosmedicine.org

Please revise your title according to PLOS Medicine's style. Your title must be nondeclarative and not a question. It should begin with main concept if possible. "Effect of" should be used only if causality can be inferred, i.e., for an RCT. Please place the study design ("A randomized controlled trial," "A retrospective study," "A modelling study," etc.) in the subtitle (ie, after a colon).

Please add line numbering to your manuscript to facilitate further review.

Please structure your abstract using the PLOS Medicine headings (Background, Methods and Findings, Conclusions).

Abstract Background: Provide the context of why the study is important. The final sentence should clearly state the study question.

Abstract Methods and Findings:

* Please ensure that all numbers presented in the abstract are present and identical to numbers presented in the main manuscript text.

Please remove the ‘What is already know on this subject’ and ‘What this study adds’ sections and replace them with an Author Summary.

Please update ‘Background’ to ‘Introduction’.

Please place citations in square brackets preceding punctuation throughout.

Please include your prospective protocol with your revised manuscript as a Supporting Information file to be published alongside your study, and cite it in the Methods section. A legend for this file should be included at the end of your manuscript. 

Please add the following statement, or similar, to the Methods: "This study is reported as per the Strengthening the Reporting of Observational Studies in Epidemiology (STROBE) guideline (S1 Checklist)."

For all observational studies, in the manuscript text, please indicate: (1) the specific hypotheses you intended to test, (2) the analytical methods by which you planned to test them, (3) the analyses you actually performed, and (4) when reported analyses differ from those that were planned, transparent explanations for differences that affect the reliability of the study's results. If a reported analysis was performed based on an interesting but unanticipated pattern in the data, please be clear that the analysis was data-driven.

Please include [preprint] in reference 12 (see our website for other reference guidelines https://journals.plos.org/plosmedicine/s/submission-guidelines#loc-references). 

The terms gender and sex are not interchangeable (as discussed in https://www.who.int/health-topics/gender); please use the appropriate term.

Please define the abbreviations in your Figures (in the caption if necessary).

You may use almost any description as the item name of your supporting information as long as it contains an "S" and number. For example, “S1 Appendix” and “S2 Appendix,” “S1 Table” and “S2 Table,” and so forth. Further information can be found at https://journals.plos.org/plosmedicine/s/supporting-information.

Please provide titles and legends for all figures (including those in Supporting Information files).

Please indicate in the figure caption the meaning of the blue shading in Figures 2-5 (and Figure S1). Please also ensure the axes are consistently labelled across all of these figures.

Please ensure your Results are consistently reported in the past tense.

Please quantify the results using 95% CIs and p values.

Tables 1 and 2: Please do not report P < 0.0001; report as P < 0.001.

Please label the two panels in Figure 4 as A and B and refer to them as such in the main text. Please also adjust the resolution of these images such that they are of comparable quality to the other figures. 

The lower panel of Figure 4 is unclear, please provide more information in a Figure legend that would allow the reader to interpret this figure without the need to refer to the main text. Please make it clear that the data used to derive this panel are from the Department of Transport and confirm that you have the appropriate permissions to reproduce these here.

Please replace the coloured vertical lines in Figure 6 with black dashed lines and make it clear that these lines denote lockdown periods. 

Please detail the analysis in the caption of Figure 6. 

In the ‘Interrupted time-series analysis’ section of the Results, the term "trend" is used to refer to a nonsignificant P value. The term trend should be used only when the test for trend has been conducted. Please revise accordingly.

In the Discussion, you include ‘as has been suggested in other international studies’ but you only cite one study. Please include further references or update the sentence as needed.

Please present and organize the Discussion as follows: a short, clear summary of the article's findings; what the study adds to existing research and where and why the results may differ from previous research; strengths and limitations of the study; implications and next steps for research, clinical practice, and/or public policy; one-paragraph conclusion.

Please remove the Contributors, Competing Interest Statement, Funding Statement, and Data availability statement sections from the main text and ensure the relevant information is captured in your answers to the Submission Form Questionnaire.

Please use the "Vancouver" style for reference formatting, and see our website for other reference guidelines https://journals.plos.org/plosmedicine/s/submission-guidelines#loc-references

Comments from the reviewers:

Reviewer #1: In this revised manuscript the authors evaluate the influence of the COVID-19 pandemic on the admission rates of major trauma in England. For this they used the TARN database, presenting with major trauma in the years 2017 to 2021. They compared the lock-down periods with the years 2018 and 2019 and evaluated excess survival and mechanisms of injury. They saw a reduction of 21% and 6,7% during the first and second lockdown respectively. This reduction was mainly in road traffic accidents, whereas the number of at home injuries of the elderly increased in the second lock down period. Moreover survival in major trauma was reduced in the first lock-down period.

Methods and statistics used are adequate. The data set is routinely gathered in a standardized well-known process. Also the survival evaluation is done in a standard, earlier published, way.

Tables and figures are adequately designed and provide a good overview of the results, however are complementary to each other.

The discussion is comprehensive and discusses the relevant questions arising form the results, concerning of the available literature.

Concerning the reduction of the likelihood of survival, of course the question arises whether there was a coincidence with a covid infection in these patients, however I presume these data are not available, but could be discussed in my opinion. This was addressed, however, in the limitations.

Limitations are adequately addressed, as well as the implications.

Reviewer #2: Dear Authors,

This is a reasonable study. It is unfortunately very sparsely referenced considering how much data is available during the last 2 years on this topic. Even a recent SRMA covered the global impact of the pandemic on major trauma epidemiology (PMID: 35723706).

This UK experience needs to be in context of other regions of the world at least in relation to the first wave, which probably the best documented. 

Reviewer #3: Thanks for the opportunity to review your manuscript. My role is as a statistical reviewer, so my review concentrates on the study design, data, and analysis that are presented. I have put general questions first, followed by queries relevant to a specific section of the manuscript (with a page/paragraph reference).

This study uses a cohort of patients that have experienced serious trauma, with data sourced from a routinely collected health data from trauma centres and units in England, compiled into a trauma registry. The main research question was whether there were changes in hospitalisations for major trauma (with a variety of outcomes examined, including volume, mechanisms, patient type, outcomes) during the first 20 months of COVID. Data was compiled into quarters for the time-series analysis of patient characteristics and care pathways, and an overall test for variation in patient characteristics (chi-square) for across the time periods was used. For changes in risk adjusted survival a weekly time series was used with a segmented regression model, testing for trend and level changes in this outcome. There is a large dataset available - across the study period there were >190 000 patients for the ITS, and >50 000 patients on the analysis of patient characteristics. 

The data and analyses presented mostly match the pre-registered protocol. The two aspects of study design I didn't see in the protocol were the selection of the comparator time periods (to the lockdown periods), and the goodness of fit test for this comparison. Were these specified after the data was available? 

I think that the application of the chi-square tests to the stratified number of admissions (Table 1) is limited. With even a moderately large sample size the chi-square test is always 'significant', e.g. the % of males in lockdown 1 vs. comparator period 1 is 54.3% vs. 55.4%, and still has p<0.05 with Bonferroni adjustment. I think interpreting the differences here on the absolute change (with CI) is much more informative than using the p-values, and I would consider just presenting the effect estimates with CI. 

One aspect of the study that I found confusing was the use of two different years for the comparator periods for lockdown 1 and lockdown 2. Was the intention to find the same length of time pre-COVID as each lockdown period to limit seasonal influences on difference between the comparator periods? It does also make it more difficult to interpret whether presence or absence of a change in lockdown 1 vs lockdown 2 is associated with differential effects of the two lockdowns vs. the different comparator periods. 

P4, Paragraph 5. Was the segmented regression model a linear regression model or another distribution? Was there any checks for residual autocorrelation? 

Figure 2 - could you add a label on the shaded time periods to indicate which lockdown period these are? And put the acronym (MTC) after the phrase in the title?

Figure 3. I would probably use 'January' if August isn't shortened. Also need a note to explain the 'RTC' acronym. 

Figure 4. Is there a way to show the traffic index on the same graph as the number of trauma patients (e.g. a secondary axis)? 

Figure 5. I wasn't sure what 'NAI' acronym was here. 

Reviewer #4: The authors ought to be congratulated for a well-done analysis across trauma patients in England. Fascinating data and well written paper. Impact of Covid on a mature well defined national trauma system and patient access to optimal care and outcomes. Well done and based on a unique database. Discussion is very well considered. I have no significant issues and enjoyed the paper and the ethical challenges defined. My only remarks concern two statements in the discussion section that I would like the authors to rethink and perhaps elaborate on. 

In the discussion the authors state that their most important finding for clinical practice is the reduction in adjusted survival in the beginning if the first lockdown. Which was equated to 2 additional deaths per 100 patients. I agree that this most certainly cannot be explained by the reduction in COVID prevalence. They state that this is likely due to reconfiguration and disruption of services, particularly for older patients presenting with low energy mechanisms of injury. Yet, the MS describes that the TARN excludes Isolated femoral neck or single pubic ramus fractures in patients > 65 years and simple isolated injuries. 

It's interesting to see the similarities and differences in trauma occurrence inflicted by imposed covid restrictions between international trauma data sets. I believe the authors have sufficiently compared and evaluated their data with studies performed in other European countries. In my opinion the most interesting statement is that the initial reduction in likelihood of survival could not be attributed to decreased care in major trauma centres or ICU. The authors support this by posing that the proportion of patients admitted to the ICU, by-passed and transferred to major trauma centres, also decreased during the second lockdown period without a corresponding significant decrease in survival. This is plausible, but the authors need to convince us that there is no independent correlation. The study from the Netherlands describes an increased trauma related mortality and risk of death associated with patients with TBI which were not admitted to the ICU. The present study does not analyze the relationship between TBI and mortality yet describes absolute reductions in moderate and severe traumatic brain injury during the second lockdown. Could this change in trauma patient survival between the first and the second lockdown be attributed to the change in TBI prevalence?

[LINK]

---

## [Decision Letter · Decision Letter 2]

12 May 2023

Dear Dr. Marincowitz,

Thank you very much for re-submitting your manuscript "Major trauma presentations and patient outcomes in English hospitals during the COVID-19 pandemic: a retrospective cohort study." (PMEDICINE-D-22-03493R2) for review by PLOS Medicine.

I have discussed the paper with my colleagues and the academic editor and it was also seen again by 3 reviewers. I am pleased to say that provided the remaining editorial and production issues are dealt with we are planning to accept the paper for publication in the journal.

[LINK]

We look forward to receiving the revised manuscript by May 19 2023 11:59PM.   

Sincerely,

Philippa Dodd, MBBS MRCP PhD

PLOS Medicine

plosmedicine.org

Requests from Editors:

GENERAL

Thank you for your detailed and considered responses to previous editor and reviewer comments. Please see below for further comments which we require that you address in full prior to publication.

Throughout, please replace the word ‘retrospective’ with ‘observational’ when describing your study.

TITLE

Please revise to read as follows, ‘Major trauma presentations and patient outcomes in English hospitals during the COVID-19 pandemic: an observational cohort study.

ABSTRACT

Line 43 – as above, please replace the word ‘retrospective’ with ‘observational’ when describing your study.

Line 45 - please include additional details of your study population including the total number of patients studied and according to each year of your study. Please also detail some of the demographic characteristics that you refer to

AUTHOR SUMMARY

Thank you for including an author summary. The author summary should consist of 2-3 succinct bullet points under each of the headings.

Suggest combining the points at lines 78 and 80 into a single statement.

Line 87 – suggest ‘appear not to have altered…’ for improved brevity

Line 85 – what do these findings mean? Authors should reflect on the new knowledge generated by the research and the implications for practice, research, policy, or public health. Authors should also consider how the interpretation of the study’s findings may be affected by the study limitations. In the final bullet point of ‘What Do These Findings Mean?’, please describe the main limitations of the study in non-technical language. Please revise in-line with this guidance.

INTRODUCTION

The introduction is very brief. We agree with reviewer #2 (please see below) that more emphasis should be placed on the current existing literature in the introduction such that the reader is not misled regarding the novelty of your study. 

Please ensure that you address past research and explain the need for and potential importance of your study. Indicate whether your study is novel and how you determined that. If there has been a systematic review of the evidence related to your study (or you have conducted one), please refer to and reference that review and indicate whether it supports the need for your study.

METHODS and RESULTS

Line 120 – as above, please replace the word ‘retrospective’ with ‘observational’ when describing your study.

Line 170 – please remove the PPI statement from the end of the methods section.

Line 195 – please ensure accurate use of parentheses it appears as though one maybe be missing and one duplicated.

Line 196 – could the low line be removed here? And the ‘.EPS’

TABLES

Table 1 – at times the use of parentheses introduces confusion – the mix of square and circular and the data split across different rows has the potential to cause problems for the reader and detract from your data. For example, age 1, row #3, currently reads, ‘-8 (-6.1%]’ should it not read ‘-8 (-6.1%)’. Similarly, age <16 row #3 currently reads, ‘[-0.4 (-0.8 to 0]’ should it not read as, ‘[-0.4 (-0.8 to 0)]’ 

In addition, we also suggest separating upper and lower CI bounds with commas rather than the word ‘to’ or hyphens (as these can be confused with the reporting of negative values) which will help to conserve space i.e. ‘[-0.4 (-0.8, 0)]’. 

We note similar issues in table 2 also. 

Please check carefully throughout, including the supporting files and revise accordingly.

FIGURES

Figure 2 – please add ‘(MTC)’ to the text below the figure so depict the abbreviation that you define.

Figure 4b – you refer to areas of shading but I cannot see any areas of shading in my version of the manuscript. Presumably this pertains to figure 4a? Please clarify/revise.

Figure 6 – what do the solid lines represent here? It appears that the dashed lines represent changes in lockdown restrictions (as per the legend). Should the graphic be adjusted, perhaps positioned horizontally? 

I was unable to access S3 Figure

*** Please revise to ensure that all detail from the figures is clearly defined for the reader in an appropriate caption affiliated to the figure such that the reader does not need to refer to the test to interpret the figures ***

REFERENCES

For in-text reference callouts please ensure that a space precedes the opening parenthesis. For example, line 113, ‘presentations[8-11].’ Should read as follows, ‘presentations [8-11].’ Please check and amend throughout where necessary.

Please also remove spaces between citations, for example line 319, ‘analysis[11, 17-19]’ should read, ‘analysis [11,17-19]’. Please check and amend throughout where relevant.

In the bibliography – please provide access dates for all web references. Please ensure that reference formatting follows our guidelines which can be found here https://journals.plos.org/plosmedicine/s/submission-guidelines#loc-references

Journal name abbreviations should be those found in the National Center for Biotechnology Information (NCBI) databases. 

SOCIAL MEDIA

To help us extend the reach of your research, please detail any Twitter handles you wish to be included when we tweet this paper (including your own, your coauthors’, your institution, funder, or lab) in the manuscript submission form when you re-submit the manuscript.

Comments from Reviewers:

Reviewer #2: Dear Authors,

This is an improved manuscript in terms of discussion. But the existing literature is still just brought up in discussion and not in introduction, which makes the reader think that this is very unique work. The introduction should document on quality existing original and review work and identify the uniqueness of this paper in that context. Some of the existing work would dwarf this labour and the authors should be upfront with it if that the case. 

Reviewer #3: Thanks for the revised manuscript and responses to my original queries. This versions addresses all the comments I had form my initial review, I don't have any further questions.

Reviewer #4: All reported remarks have been amended appropriately. 

I have no further remarks. 

Great article.

[LINK]

---

## [Editor Report · Decision Letter 3]

19 May 2023

Dear Dr Marincowitz, 

On behalf of my colleagues and the Academic Editor, Dr. Martin Schreiber, I am pleased to inform you that we have agreed to publish your manuscript "Major trauma presentations and patient outcomes in English hospitals during the COVID-19 pandemic: an observational cohort study." (PMEDICINE-D-22-03493R3) in PLOS Medicine.

Prior to publication, please remove all sub-headings from the discussion (including ‘conclusions’) such that it reads as a single piece of continuous prose.

PRESS

Sincerely, 

Philippa Dodd, MBBS MRCP PhD 

PLOS Medicine